# Environmentally-Benign Phytic Acid-Based Multilayer Coating for Flame Retardant Cotton

**DOI:** 10.3390/ma13235492

**Published:** 2020-12-02

**Authors:** Eva Magovac, Igor Jordanov, Jaime C. Grunlan, Sandra Bischof

**Affiliations:** 1Faculty of Textile Technology, University of Zagreb, HR-10000 Zagreb, Croatia; emagovac@ttf.hr; 2Faculty of Technology and Metallurgy, Ss. Cyril and Methodius University in Skopje, 1000 Skopje, Republic of North Macedonia; jordanov@tmf.ukim.edu.mk; 3Department of Mechanical Engineering, Texas A&M University, College Station, TX 77840, USA

**Keywords:** layer-by-layer assembly, flame retardant, cotton, phytic acid, chitosan, urea

## Abstract

Chemically bleached cotton fabric was treated with phytic acid (PA), chitosan (CH) and urea by means of layer-by-layer (LbL) deposition to impart flame retardant (FR) behavior using only benign and renewable molecules. Samples were treated with 8, 10, 12 and 15 bilayers (BL) of anionic PA and cationic CH, with urea mixed into the aqueous CH solution. Flammability was evaluated by measuring limiting oxygen index (LOI) and through vertical flame testing. LOI values are comparable to those obtained with commercial flame-retardant finishes, and applying 10 or more bilayers renders cotton self-extinguishing and able to pass the vertical flame test. Microscale combustion calorimeter (MCC) measurements show the average reduction of peak heat release rate (pHRR) of all treated fabrics of ~61% and the reduction of total heat release (THR) of ~74%, in comparison to untreated cotton. Decomposition temperatures peaks (T_1max_) measured by thermogravimetric analyzer (TG) decreased by approximately 62 °C, while an average residue at 650 °C is ~21% for 10 and more bilayers. Images of post-burn char indicate that PA/CH-urea treatment is intumescent. The ability to deposit such a safe and effective FR treatment, with relatively few layers, makes LbL an alternative to current commercial treatments.

## 1. Introduction

Cotton is one of the best-selling textiles in the world, used for a wide range of products such as sportswear, fashion garments as well as protective clothing due to its softness and moisture absorption, which makes it comfortable to wear. This absorption is enabled by hydroxyl groups in the cellulose molecule that attract water and make it hydrophilic. However, high flammability is the primary undesirable property of cotton, making it inappropriate for protective clothing and safety workwear requiring fire safety. To reduce flammability, cotton has been treated with commercially available flame-retardant finishes based on halogen, organo-halogen, antimony organo-halogen or organophosphorus chemistries [1]. Many of these flame retardants are toxic to humans, as well as to the environment, causing endocrine disruption, infertility, cancer, neurobehavioral problems, as well as embryotoxic and teratogenic effects [2]. Inhalation of toxic volatile products generated in a fire (e.g., carbon monoxide, hydrogen cyanide, hydrocarbons, dioxin, acrolein, formaldehyde, etc.) can lead to death. Until recently, flame retardants based on organophosphorus compounds have been considered safe, but in recent studies they have been found to be persistent in the atmosphere, soil, water and in biological samples [3].

The most durable flame retardant (FR) finishes for cotton fabric are typically organophosphorus compounds based on tetrakis (hydroxymethyl) phosphonium derivatives and N-methyloldimethyl phosphonopropioamide applied by a pad-dry-cure process, which produces toxic smoke during curing [4]. Formaldehyde is one of the toxic compounds released in these finishing processes. Formaldehyde-free alternatives for cotton finishing are polycarboxylic acid-based flame retardants, such as 1,2,3,4-butanetetracarboxylic acid (BTCA), succinic acid (SA), citric acid and malic acid (MA) [5,6]. Layer-by-layer (LbL) deposition of more environmentally-benign compounds based on phosphorus-nitrogen synergy has emerged as another promising approach for cotton flame retardancy [7,8]. This aqueous treatment consists of polyanion and polycation solution exposure, resulting in a multilayer nanocoating [9,10]. This versatile technique is applicable to nearly any surface and can make use of polymers, nanoparticles and various small molecules. 

Chemically bleached cotton fibers are generally negatively-charged due to the presence of carboxyl and hydroxyl-groups [11]. However, cotton cellulose modified by quaternary ammonium compounds that block anionic groups results in a positive charge [12]. Charged surface of cotton fibers make them an ideal substrate for the LbL deposition [13]. LbL treatment of cotton involves dipping/immersing of fabric into the oppositely-charged polyelectrolyte solutions or simply spraying with polyelectrolyte solutions [14]. Repeated exposure to oppositely-charged polyelectrolytes can be used to deposit bilayers (BL), trilayers (TL) or quadlayers (QL) with a desired functionality, such as combination of hydrophobicity-flame retardancy-conductivity [15], hydrophobicity-flame retardancy [16,17], or antimicrobial-flame retardancy [18,19]. 

In the present study, three environmentally-benign compounds have been used: phytic acid (PA), chitosan (CH) and urea, where negatively-charged PA and positively-charged CH are known to form an FR nanocoating on cotton [20]. Phytic acid stores phosphorus in plants [21], while chitosan is a linear polysaccharide produced commercially by deacetylation of the chitin shells in the crustaceans [22]. For FR purposes, chitosan acts as a source of nitrogen as well as blowing agent [23]. Urea is the principal end product of metabolism in mammals and provides an additional source of nitrogen [24]. Thirty bilayers of CH/PA has been shown to impart self-extinguishing behavior to cotton [20]. Thirty bilayers can be reduced to 4 BL by adding divalent metal ions such as barium into CH [25]. Here it is shown that the addition of nitrogen-rich urea reduces the number of bilayers from 30 to 10 for the same level of flame retardancy saving the energy by skipping drying at 80 °C after each immersing/padding step. Additionally, this 10 BL nanocoating increases the LOI of cotton from 18 to 28%. Accomplishing such effective fire protection using only environmentally-benign chemistries and relatively few processing steps makes this a very scalable treatment. 

## 2. Materials and Methods

Chemically bleached, desized cotton fabric, with a weight of 119 g/m^2^, was supplied by the USDA Southern Regional Research Center (New Orleans, LA, USA). Branched polyethyleneimine (BPEI, M = 25,000 g/mol, ≤1% water), phytic acid sodium salt hydrate (PA), urea, hydrochloric acid (HCl) and sodium hydroxide-pellets (NaOH), were all purchased from Sigma Aldrich (Milwaukee, WI, USA). Chitosan (CH) powder (M ~ 60,000 g/mol 75−85% deacetylated) was purchased from G.T.C. Bio Corporation (Qingdao, China). For the preparation of all polyelectrolyte solutions, 18.2 mΩ deionized (DI) water was used.

In an effort to improve coating adhesion to cotton, an aqueous cationic solution of BPEI (5 wt%) was prepared and the textile was immersed to deposit a primer layer. An aqueous anionic solution of PA (2 wt%) and cationic CH (0.5 wt%), both prepared with DI, were magnetically stirred for 24 h. Urea (10 wt%) was added to the CH solution and magnetically stirred until completely dissolved. The pH of all of these solutions was adjusted to 4, with 1 M NaOH or 1 M HCl, just before LbL deposition. Cotton was washed in a standard detergent solution, dried in an oven for 24 h at 80 °C and cut into five 3 inch × 12 inch samples. Four cotton samples were alternately immersed in the PA and CH-urea solutions, depositing 8, 10, 12 and 15 BL, as shown in Table 1. The remaining cotton sample was left untreated as a control. The process of LbL deposition is shown in Figure 1. 

The immersion time is 5 min for the first layer and 1 min for each additional layer. Each immersion step is followed by rinsing in DI water. All samples are dried in the oven at 80 °C for 24 h after the LbL treatment. 

The weights of all samples were measured after drying in the oven at 80 °C for 24 h, before and after LbL treatment, to calculate weight gain (%) using the following equation:(1)$$\mathrm{weight}\;\mathrm{gain}\;\left(\%\right)\;=\;\frac{\;m\;\left(\mathrm{treated}\right)-m(\mathrm{untreated})}{m\;(\mathrm{untreated})}\;\times \;100$$

Limiting oxygen index (Dynisco, Heilbronn, Germany) was measured according to ISO 4589-2:2017 [26]. 

Vertical flame testing was carried out in a standard chamber (Govmark, Farmingdale, NY, USA) according to ASTM D6413/D6413M-15 [27]. 

A Govmark MCC-2 (Heilbronn, Germany) microscale combustion calorimeter (MCC) was used to measure heat release of cotton samples according to ASTM D7309-19a [28]. A temperature range of 75–650 °C was used with the mixture of gases consisting of 20% N_2_ and 80% O_2_, 100 mL/min gas flow rate) and heating rate of 1 °C/min. The repeatability of the MCC measurement was determined on three replicated samples for each treatment with standard deviations. 

Thermogravimetric analysis (TGA) was performed with a PerkinElmer Pyris 1 (Shelton, CT, USA). All samples were heated from 50 to 850 °C with a heating rate of 30 °C/min in air (flow rate: 30 mL/min). The morphology of all samples, as well as post-burn char, was imaged using a Tescan MIRA\\LMU FE-SEM (Scanning Electron Microscope, SE detector, 5 kV, Brno, Czech Republic). All samples were coated with 5 nm of chromium for better conductivity (Q150T ES Sputter Coater, Quorum Technologies, Laughton, UK), with the exception of the char samples.

## 3. Results and Discussion

Cotton samples were treated with a varying number of chitosan-urea/phytic acid bilayers. The tendency toward linear weight gain can be seen in Table 1.

The minimum oxygen fraction in an O_2_/N_2_ mixture that supports combustion, as well as the time required to complete combustion, was measured for these treated samples. LOI for untreated cotton is between 18% and 19%, depending on the type of fabrics, construction, weight, moisture, ambient temperature, etc. For cotton treated with commercial durable flame retardants, the LOI is usually within the range of 28–29% [29]. Here, LOI increases with the number of CH-urea/PA bilayers deposited. Cotton treated with 8 BL has an LOI of 26%, which increases to 31% with 15 BL. Cotton treated with 10, 12 and 15 BL satisfy commercial requirements, with the values of 28% or above. Burning time of samples treated with 10, 12, 15 BL is ~35 s, which means that samples burn more slowly than untreated cotton.

The results of vertical flame testing correlate with the LOI results. Samples treated with 10, 12 and 15 BL pass the vertical flame test (VFT) with char lengths between 12.0 and 13.0 cm, with no afterflame and afterglow time, as summarized in Table 2.

Figure 2 shows microscale combustion calorimetry (MCC) heat release rates (pHRR) as a function of temperature for cotton LbL-treated with varying bilayers of PA/CH-urea.

Factors influencing pHRR curves are homogeneity, sample weight, flow rate perturbation, oxygen level and loading/types of additives [30,31]. Three different groups of curves are observed: untreated cotton (control), cotton treated with 8 BL and cotton treated with 10, 12 or 15 BL. These results correlate well with VFT results. Table 3 summarizes the MCC results.

Peak release rates (pHRR) of untreated cotton is 234.8 W/g, while total heat release (THR) is 11.1 kJ/g at ~380 °C. pHRR values for cellulose materials vary depending on their chemical composition (content of lignin, hemocellulose, impurities, etc.). Peak heat release rate of FR treated cotton is decreased by 50%, compared to untreated cotton. Among all treated samples, 8 BL exhibits the lowest reduction of pHRR (57.0%) and THR (67.6%), which correlates with the results of vertical flame testing (where the 8 BL sample burns completely). Cotton treated with 10, 12 or 15 BL reduces pHRR by ~61% and THR ~74%. In all cases, pHRR, THR and T_pHRR_ values decrease steadily with increasing bilayers deposited. 

Figure 3 shows TGA curves of untreated and LbL-treated cotton samples with three stages of weight loss, while Figure 4 shows derivative weight as a function of temperature for all samples. 

The first weight loss starts between 50 and 100 °C, due to evaporation of moisture, and is identical for untreated and treated samples. The first decomposition stage begins between 250 and 400 °C, with dehydration and depolymerization. At this stage, T_1max_ cellulose loses 95% of its weight, generating non-flammable gases, primary char residue and levoglucosane. All LbL-treated cotton samples exhibit a shift to lower temperatures, by ~60 °C. The curves differ more dramatically at the second decomoposition stage, which starts between 500 and 650 °C. Its maximum is T_2max_, where levoglucosane produces flammable gases and secondary char [32]. As shown in Table 4, the highest mass loss of untreated cotton (59%) appears at ~389 °C, while at ~585 °C it loses 95% of its mass.

The char consists of impurities or inorganic compounds, making up 1–5% of untreated cotton. On the other hand, T_1max_ of LbL-treated cotton appears at ~327 °C, with an average mass loss of 37%. This means that the CH-urea/PA coating decreases the decomposition temperature of cotton. At this stage non-flammable gases generate that dilute the concentration of the combustible gases and absorb heat causing bubbling [33]. At the same time, urea catalyzes the reaction of PA as well as the decomposition of cellulose at low temperature, thus forming intumescent char, which acts as physical barrier that blocks heat and oxygen [34]. At T_2max_ the average mass loss for all treated samples is around 80 %. At 800 °C the oxidation of all organic compounds occurs. At 650 °C, the 8 BL coated cotton has a char of ~7%, while the char yield of 10, 12 and 15 BL is ~14%.

Figure 5 shows SEM micrographs of untreated and LbL-treated cotton fabric. Untreated cotton fabric’s smooth surface (Figure 5a) contrasts with the rough, uneven and paste-like surface of the treated samples (Figure 5b–e).

These images confirm that CH-urea/PA was successfully deposited onto the fabric and visually the thickness increases with the number of bilayers deposited. SEM images of char after performing vertical flame testing are shown in Figure 6.

Untreated cotton completely combusts, so no char residue was obtained. All of the charred fabric samples reveal a bubbling effect, which is characteristic of intumescent flame-retardant systems. Chitosan acts as a carbon donor, phytic acid as an acid donor and low-molecular weight urea as a blowing agent that generates gas. The foamed char acts as a physical barrier to slow heat and mass transfer between the gas and condensed phases [35].

## 4. Conclusions

Cotton fabric was successfully treated with an environmentally-benign multilayer nanocoating to reduce flammability. Layer-by-layer deposition of chitosan-urea and phytic acid solutions produced this effective intumescent treatment. LOI of cotton coated with 10, 12 and 15 BL is in the range 28–31%, confirming that LbL-treated fabric is comparable to commercially available cotton flame retardant finishes. 10 BL of CH-urea/PA applied to cotton passes the standard vertical flame test. The average reduction of pHRR of all treated fabrics is 61% and the reduction of THR, in comparison to untreated cotton, is 74%. TGA reveals an average residue at 650 °C of ~14% for 10, 12 and 15 BL, confirming the intumescent effectiveness of the treated cotton. SEM images of post-burn char show characteristic intumescent bubbles. This work demonstrates that the number of CH/PA bilayers can be dramatically reduced to 10 BL, by adding urea to the CH solution, which makes this treatment much easier to process. The ability to deposit such a safe and effective FR treatment, makes LbL an ecologically-friendly alternative to current commercial treatments.

## Figures and Tables

**Figure 1 materials-13-05492-f001:**
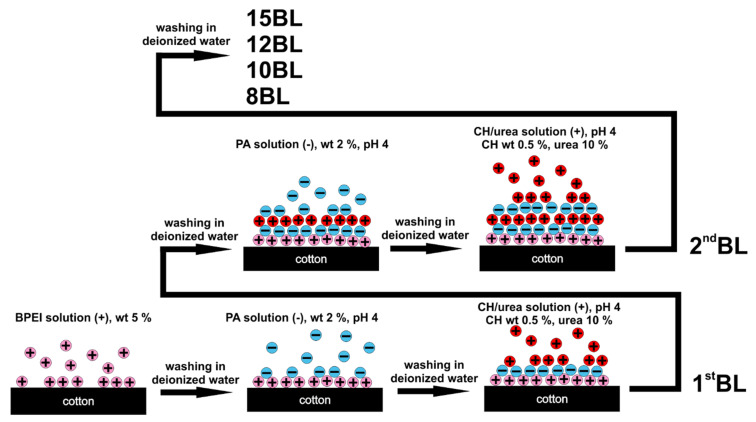
Schematic of layer-by-layer deposition of flame retardant nanocoatings on cotton fabric.

**Figure 2 materials-13-05492-f002:**
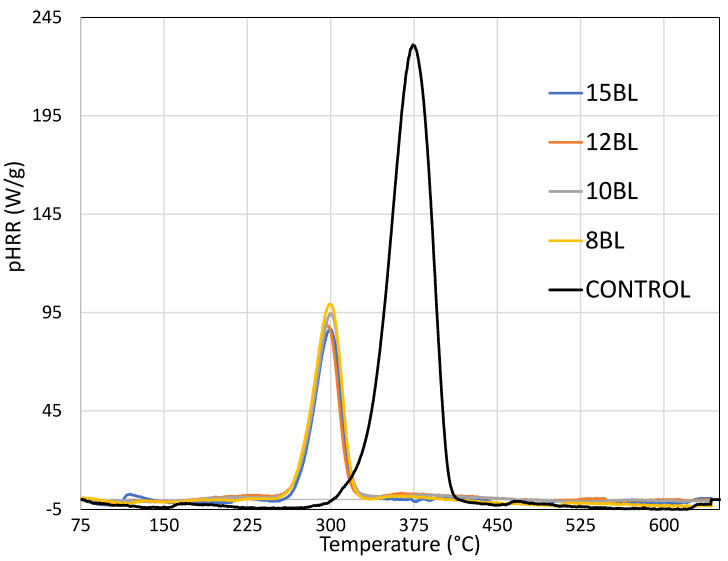
Heat release rates as a function of temperature for cotton treated with varying bilayers (8 BL-15 BL) of PA/CH-urea.

**Figure 3 materials-13-05492-f003:**
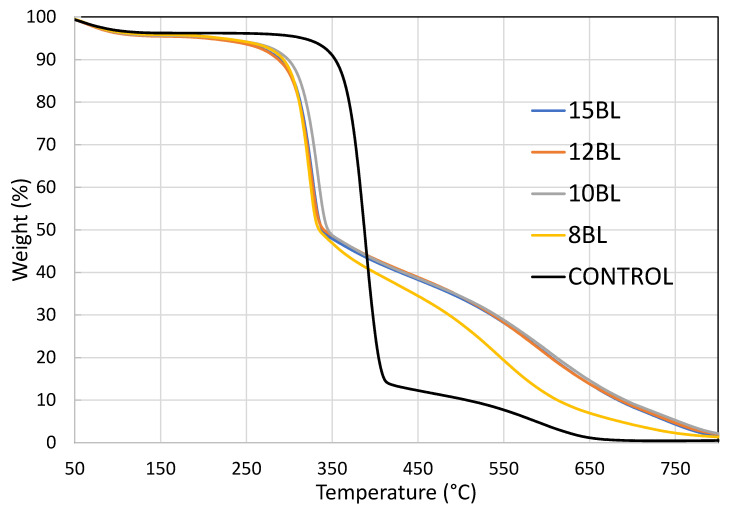
Cotton weight as a function of temperature for untreated and LbL-treated samples.

**Figure 4 materials-13-05492-f004:**
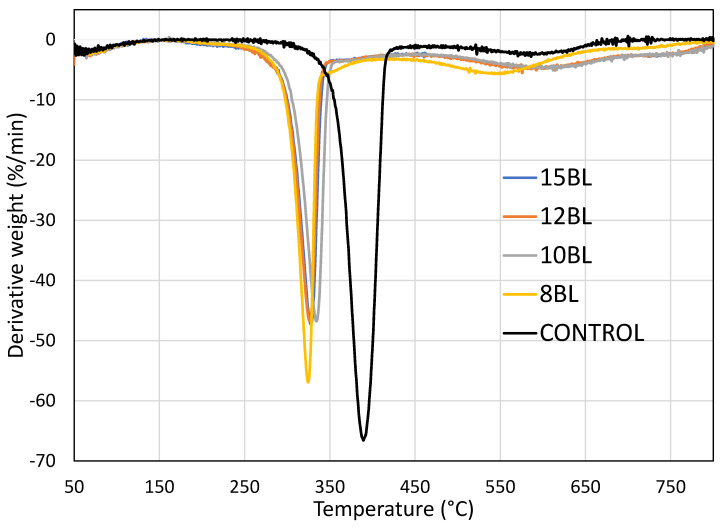
Derivative weight as a function of temperature for untreated and LbL-treated samples.

**Figure 5 materials-13-05492-f005:**
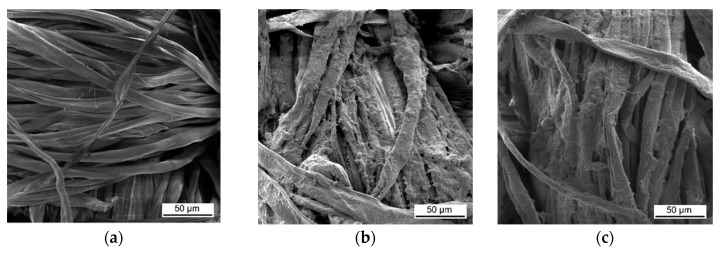
SEM images of (**a**) untreated cotton and (**b**) cotton treated with 8, (**c**) 10, (**d**) 12, and (**e**) 15 bilayers (BL) chitosan (CH)-urea/phytic acid (PA).

**Figure 6 materials-13-05492-f006:**
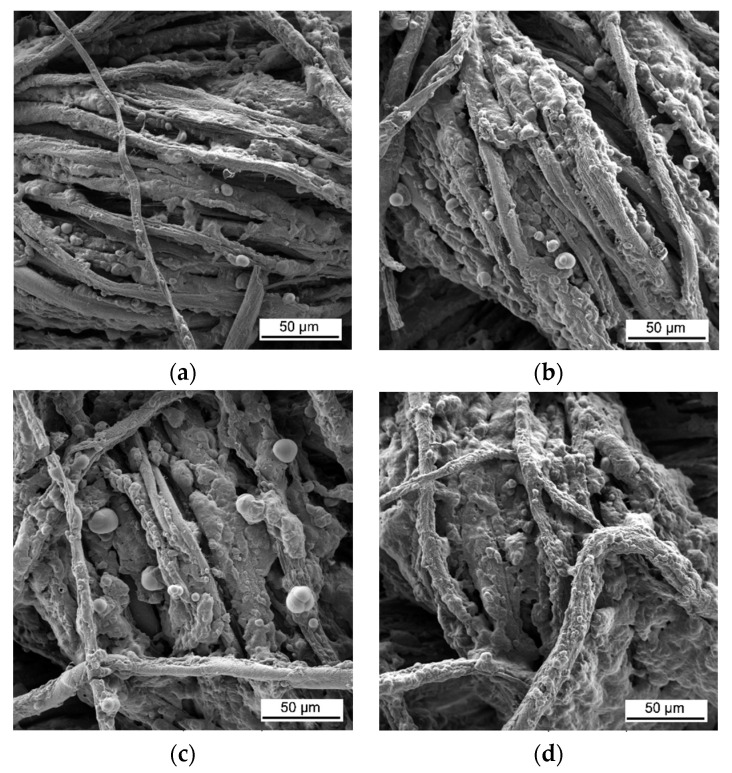
SEM images of the char residue of treated cotton after performing vertical flame testing: (**a**) 8, (**b**) 10, (**c**) 12, and (**d**) 15 BL.

**Table 1 materials-13-05492-t001:** Weight gain and limiting oxygen index data.

Number of BL	Weight Gain (%)	LOI (%)	Time (s)
Control	-	18	25
8	12.36	26	26
10	17.29	28	40
12	18.19	29	34
15	20.12	31	30

**Table 2 materials-13-05492-t002:** Vertical flame test results for cotton samples treated with varying number of bilayers.

Number of BL	8	10	12	15
	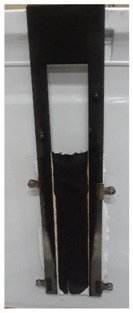	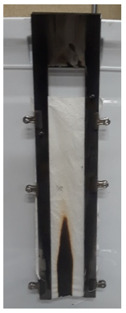	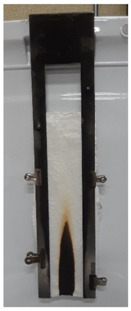	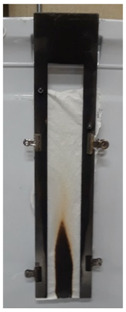
Char length (cm)	n/a	13.0	12.0	12.5
After flame time (s)	n/a	0	0	0
After glow time (s)	n/a	0	0	0

**Table 3 materials-13-05492-t003:** Heat release values of layer-by-layer (LbL) treated cotton fabric (with standard deviations).

Number of BL	pHRR (W/g)	∆HRR (%)	THR (kJ/g)	∆THR (%)	T_pHRR_ (°C)
Control	234.8 (5.7)	-	11.1 (0.9)	-	380 (1.7)
8 BL	101.0 (4.8)	57.0	3.6 (0.8)	67.6	302 (2.3)
10 BL	95.1 (6.3)	59.5	3.3 (0.7)	70.3	303 (3.0)
12 BL	88.6 (4.5)	62.3	3.0 (0.5)	73.0	299 (1.8)
15 BL	86.2 (6.1)	63.3	2.2 (0.8)	80.2	303 (2.1)

**Table 4 materials-13-05492-t004:** Summary of thermogravimetric analysis of treated and untreated cotton.

Number of BL	T_1max_ (°C)	Char Yield at T_1max_ (%)	T_2max_ (°C)	Char Yield at T_2max_ (%)	Char Yield at 650 °C (%)
Control	389	46.28	585	5.20	1.17
8	323	63.53	552	19.10	6.96
10	334	62.50	604	21.13	14.63
12	326	62.63	592	21.93	13.82
15	327	63.00	607	19.76	13.80

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
