# Peer review of "Environmentally-Benign Phytic Acid-Based Multilayer Coating for Flame Retardant Cotton"

_materials, 2020, doi:10.3390/ma13235492_

Round 1

Reviewer 1 Report

Manuscript is well written, it presents very interesting topic of LbyL nanocoating for FR application

Author Response

We are grateful for the recommendation for publishing without any changes.

Reviewer 2 Report

Magovac et al. treated cotton fabric with phytic acid (PA), chitosan (CH) and urea by means of layer-by-layer (LbL) deposition to impart flame retardant (FR) behavior. The thermal decomposition and flame retardant properties were studied. However, there are some issues should be addressed before publication. Here are the detailed comments:

(1) The introduction needs to be improved. There are some reports flame retardant cotton fabrics using phytic acid (PA), chitosan (CH): International Journal of Biological Macromolecules 2019, 140, 303-310; Carbohydrate Polymers 2020, 237, 116173. Compared to these literatures, the authors should state the advances of this work.

(2) In Table 1, what is the meaning of time?

(3) In Figure 2, there is no peak for 10BL. Please check it.

(4) In Table 3, please add the error scale for MCC data.

(5) The authors declared “At 650 °C, the 8 BL coated cotton has a char of ~ 7 %, while the char yield of 10, 12 and 15 BL is ~ 21 %.”, which is not consistent with Fig. 3 and Table 4. From Fig. 3, it can be seen that the char yield of 10, 12 and 15 BL at 650 °C is below 10%. Please check it.

(6) Washing durability is important for flame retardant fabrics. Please add the washing durability test for the treated cotton fabrics in this work.

Author Response

Answers on the reviewers comments:

(1) The introduction needs to be improved. There are some reports flame retardant cotton fabrics using phytic acid (PA), chitosan (CH): International Journal of Biological Macromolecules 2019, 140, 303-310; Compared to these literatures, the authors should state the advances of this work.

Line 67, The advances of this work were added to Introduction: „30 BL of CH/PA has been shown to impart self-extinguishing behaviour to cotton [20]. 30 BL can be reduced to 4 BL by adding divalent metal ions such as barium into CH [25]. Here it is shown that the addition of nitrogen-rich urea reduces the number of bilayers from 30 to 10 for the same level of flame retardancy saving the energy by skipping drying at 80 °C after each immersing/padding step. Additionally, this 10 BL nanocoating increases the LOI of cotton from 18 to 28 %.“

Carbohydrate Polymers 2020, 237, 116173.

Line 60 The reference to this article was added as example of multifunctional properties of cotton (FR-antimicrobial) by LbL deposition.

„Repeated exposure to oppositely-charged polyelectrolytes can be used to deposit bilayers (BL), trilayers (TL) or quadlayers (QL) with a desired functionality, such as combination of hydrophobicity – flame retardancy – conductivity [15], hydrophobicity – flame retardancy [16,17], or antimicrobial – flame retadancy [18,19].“

(2) In Table 1, what is the meaning of time?

Line 130, Added: „Burning time of samples treated with 10, 12, 15 BL is ~ 35 s, which means that samples burn more slowly than untreated cotton.“

(3) In Figure 2, there is no peak for 10BL. Please check it.

Line 141, Figure 2 has been changed, 10 BL curve is added. Thank you very much for pointing this out!

(4) In Table 3, please add the error scale for MCC data.

In Table 3 heat release values of LbL treated cotton fabric (with standard deviation) have been added.

(5) The authors declared “At 650 °C, the 8 BL coated cotton has a char of ~ 7 %, while the char yield of 10, 12 and 15 BL is ~ 21 %.”, which is not consistent with Fig. 3 and Table 4. From Fig. 3, it can be seen that the char yield of 10, 12 and 15 BL at 650 °C is below 10%. Please check it.

Line 160, figure 3 has been changed.

Line 181, the text has been changed:At 650 °C, the 8 BL coated cotton has a char of ~ 7 %, while the char yield of 10, 12 and 15 BL is ~ 14 %.“

(6) Washing durability is important for flame retardant fabrics. Please add the washing durability test for the treated cotton fabrics in this work.

Our future studies will be dealing with different curing options to achieve the washing durability of LbL treated cotton fabric. We are grateful for the suggestion.

Reviewer 3 Report

This work is foundational but has practical significance, bio-based FRs have the extensive prospect, some suggestions are given as following.

  1. The word in the title “nanocoating”, but I don’t find any characterization for nanocoating in the text, so the title is not proper.
  2. Pay attention to the unit, “g/m2” inline 73 is not standard.
  3. For the MCC test, “with a mixture of N2/O2 (20 % /80 %, 100 ml/min gas flow rate)”, do you mean the mixture gases consist of 20% N2 and 80% O2? please to confirm.
  4. So many papers for flame retardant cotton fabric by LBL,CS and PA are also common FRs, what do you think are your highlights?
  5. The data in table 4 is inconsistent with Fig. 3,especially for the char yield value at 650℃, please make a correction.

Author Response

Answer on Comments and Suggestions for Authors:

This work is foundational but has practical significance, bio-based FRs have the extensive prospect, some suggestions are given as following.

1. The word in the title “nanocoating”, but I don’t find any characterization for nanocoating in the text, so the title is not proper.

Line 3, the title has been changed to Environmentally-benign phytic acid based multilayer coating for flame retardant cotton

2. Pay attention to the unit, “g/m2” inline 73 is not standard.

Line 75, the unit has been changed: „Chemically bleached, desized cotton fabric, with a weight of 119 g/m2, was supplied by the...“

3. For the MCC test, “with a mixture of N2/O2 (20 % /80 %, 100 ml/min gas flow rate)”, do you mean the mixture gases consist of 20% N2 and 80% O2? please to confirm.

Line 106, A temperature range of 75 – 650 °C was used with the mixture of gases consisting of 20 % N2 and 80 % O2, 100 ml/min gas flow rate) and heating rate of 1 °C/min.

4. So many papers for flame retardant cotton fabric by LBL,CS and PA are also common FRs, what do you think are your highlights?

New text has been added at Line 67:

There are many papers dealing with flame retardancy of cotton treated with environmentally-benign compounds such as CS and PA by means of LbL deposition, but the highlight of this paper is adding urea into CH to reduce the number of BL from 30 to only 10 retaining saisfactory commercial requirements, with the values of 28 % or above.

5. The data in table 4 is inconsistent with Fig. 3,especially for the char yield value at 650℃, please make a correction.

Line 160, figure 3 has been changed.

Line 181, the text has been changed:At 650 °C, the 8 BL coated cotton has a char of ~ 7 %, while the char yield of 10, 12 and 15 BL is ~ 14 %.“

We are grateful for your notification of our mistakes which helped us to improve the papers quality!